# Plasticity without dislocations in a polycrystalline intermetallic

Hubin Luo[1,2], Hongwei Sheng[3,4], Hongliang Zhang [2], Fengqing Wang[1], Jinkui Fan[1], Juan Du [1], J. Ping Liu[1,5] & Izabela Szlufarska[2]

Dislocation activity is critical to ductility and the mechanical strength of metals. Dislocations are the primary drivers of plastic deformation, and their interactions with each other and with other microstructural features such as grain boundaries (GBs) lead to strengthening of metals. In general, suppressing dislocation activity leads to brittleness of polycrystalline materials. Here, we find an intermetallic that can accommodate large plastic strain without the help of dislocations. For small grain sizes, the primary deformation mechanism is GB sliding, whereas for larger grain sizes the material deforms by direct amorphization along shear planes. The unusual deformation mechanisms lead to the absence of traditional Hall-Petch (HP) relation commonly observed in metals and to an extended regime of strength weakening with grain refinement, referred to as the inverse HP relation. The results are first predicted in simulations and then confirmed experimentally.

[1] Key Laboratory of Magnetic Materials and Devices, Ningbo Institute of Materials Technology and Engineering, Chinese Academy of Sciences, Ningbo 315201, China. [2] Department of Materials Science and Engineering, University of Wisconsin, Madison, WI 53706-1595, USA. [3] Department of Physics and Astronomy, George Mason University, Fairfax 22030, USA. [4] Center for High Pressure Science and Technology Advanced Research, Shanghai 201203, China. [5] Department of Physics, University of Texas at Arlington, Arlington, TX 76019, USA. Correspondence and requests for materials should be addressed to J.P.L. (email: pliu@uta.edu) or to I.S. (email: szlufarska@wisc.edu)

Grain size engineering is one of the most important ways to strengthen a metallic system. The smaller the grain size, the higher the strength. This phenomenon, referred to as the Hall–Petch (HP) relation[1,2], arises from blocking of dislocation motion by the grain boundaries (GBs). HP relation has been a part of the textbooks since the 1960s and it has been observed in many metallic systems, including both elemental and multi-component metals[3–6]. In the past couple of decades, it has been established that the scaling of strength with grain size described by HP relation may breakdown in face centered cubic (fcc) metals and even lead to weakening of the materials with continued grain refinement when the grain size is below ~12–15 nm[7–13]. This breakdown is often referred to as the inverse HP relation and it has been attributed to the dominant role played by GBs in strain accommodation in the small grain size regime. In non-fcc metallic systems, the inverse HP relation has been previously predicted by molecular dynamics (MD) simulations[14–16], but it has been generally absent in experiments[3,5,6,17–19]. The likely reason is that grain size weakening (if present) is expected only for very small grain sizes, smaller than typically explored in experiments. Regardless of whether inverse HP relation is present in the small grain size regime, for larger grain sizes most metals exhibit the traditional grain size strengthening, consistent with the HP relation.

While blocking of dislocation motion increases the strength of metals, it generally occurs at the cost of ductility[20]. Materials where dislocation motion is energetically unfavorable and therefore suppressed are usually brittle because dislocations are the primary drivers of plasticity[21]. In this regard, exploring strain accommodation mechanisms other than dislocation could open up opportunities for design of materials with outstanding mechanical properties since the different mechanisms of plasticity could be controlled independently, e.g., by composition and microstructure.

Here, we investigate $SmCo_5$, which is an intermetallic with a space group P6/mmm (hexagonal but not close-packed). We demonstrate by MD simulations that this material can undergo large plastic deformation without the aid of dislocations. By investigating grain size dependence of strength, we reveal that $SmCo_5$ is characterized by a clear and large regime of the inverse HP relation in contrast to typical observations reported for metallic systems[7,8].

## Results

**Deformation behavior.** Simulations are carried out using the embedded atom method (EAM) potential, fitted to multiple properties of Sm, Co, and Sm–Co compounds (see Supplementary Note 1, Tables 1–3 and Figs. 1–4). We investigate samples with grain sizes ranging from 5 to 65 nm in diameter. Each sample contains ten grains with random grain orientations. From sample to sample, grain size changes but the grain orientations are kept the same (see Methods). We have performed simulations of uniaxial tension and of uniaxial compression at the strain rate of $10^8 \, s^{-1}$. Qualitative trends of strength with grain size and deformation mechanisms are the same between tensile and compressive loading (Supplementary Fig. 5). In Fig. 1a, b, we show the results of simulated compression. Surprisingly, we find that grain size softening persists up to grain diameters as large as 37 nm. For larger grains, the flow stress plateaus with no sign of trend reversal (i.e., no traditional HP relation). For comparison we show the results of analogous (tensile) simulations for Cu[9], where there is a clear regime of traditional HP strengthening and a limited inverse HP regime for grain sizes smaller than ~12 nm. $SmCo_5$ samples exhibit clear plastic deformation for all grain sizes without formation of voids or cracks, even if the true strain is as large as 18%.

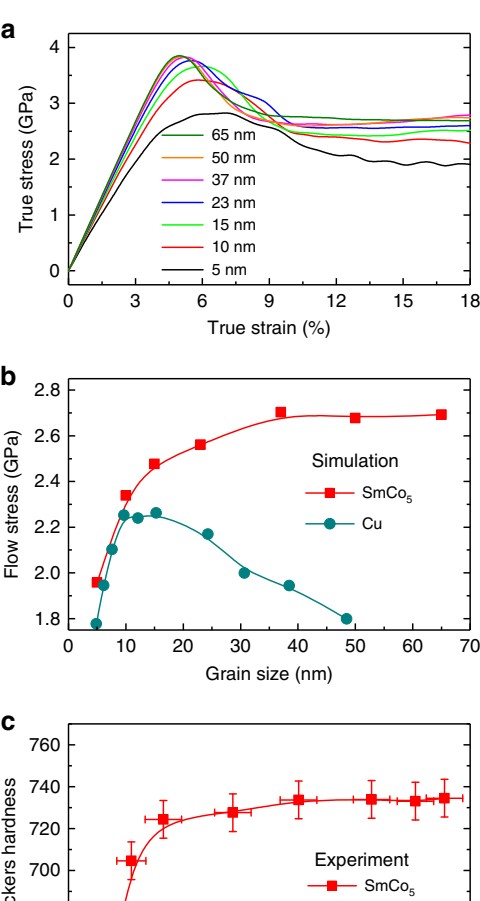

**Fig. 1** Mechanical behavior of $SmCo_5$ as a function of grain size. **a** Simulated stress–strain curve of $SmCo_5$ with different grain sizes under uniaxial compression at strain rate of $10^8 \, s^{-1}$. **b** Flow stresses of simulated $SmCo_5$ and Cu. Data for Cu are results of tensile simulation at a strain rate of $5 \times 10^8 \, s^{-1}$ and are adapted from ref. [9]. The flow stress for $SmCo_5$ is calculated from (**a**) for strains larger than 10.5%. **c** Experimentally measured Vickers hardness of $SmCo_5$ as a function of grain size (see Methods). The error bars represent deviations of grain size and hardness from their averages

To verify predictions from our simulations, we performed measurements of microhardness of experimental $SmCo_5$ samples with different grain sizes. Details are provided in Methods and Supplementary Fig. 6. We found that there is an excellent agreement between the experimentally measured dependence of hardness on grain size (Fig. 1c) and the dependence of strength on grain size predicted by MD simulations (Fig. 1b). Since microhardness and strength are expected to be roughly proportional to each other, the experiments support predictions from our simulations. If we approximate the strength to be one third of the Vickers hardness, we can find that our experimentally measured strength is comparable to the flow strengths of $SmCo_5$ obtained from MD simulations. Besides, $SmCo_5$ has also a comparable strength to that of hexagonal closed packed (hcp)-Co (~2 GPa) at the grain size of 12 nm[22].

**PES and incomplete slip.** In fcc and hcp metals, GB strengthening with decreasing grain size has been attributed to the

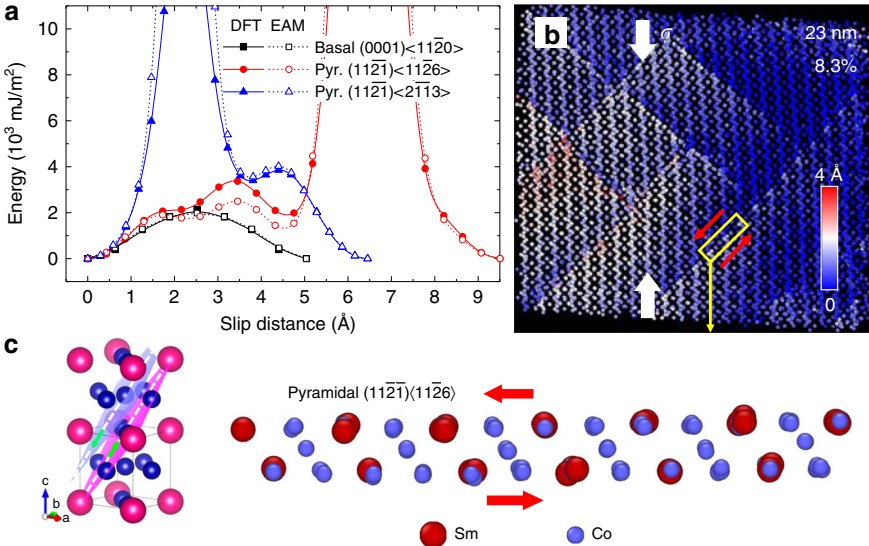

**Fig. 2** Incomplete dislocation slip in SmCo₅. **a** Potential energy surfaces (PESs) of basal and pyramidal slips calculated by using DFT and EAM potential. **b** Pyramidal $2c + a$ slip $[(11\bar{2}\bar{1})\langle11\bar{2}6\rangle]$ in a grain of the sample with a grain diameter of 23 nm under compressive strain of 8.3%. Atoms are colored by their displacements using the unstrained structure as the reference. White arrows represent external stress and red arrows show the direction of the resolved shear stress. **c** Atomic-level view of the pyramidal $2c + a$ slip

reduction of the number of dislocations piling up at GBs in small-grained materials[5,23]. In our simulations of SmCo₅, we have not observed any complete slip of dislocations. This observation is consistent with the relatively high-energy barriers for dislocation nucleation and motion found in our calculations of the potential energy surface (PES) of rigid slip. In Fig. 2a, we show examples of PES calculated both with the density functional theory (DFT) method and with the EAM potential. The energy maximum (EM) on the PES is related to the energetics of dislocation movement[24,25]. One should note that there is an excellent agreement between the results produced by the two methods, despite the fact that the potential was not fitted directly to PES (Supplementary Note 1). The agreement thus shows that the EAM potential is capable of correctly capturing dislocation physics in SmCo₅. The lowest EM (1982 mJ/m² from EAM) occurs for the basal slip $[(0001)\langle11\bar{2}0\rangle]$. The pyramidal $2c + a$ slip $[(11\bar{2}\bar{1})\langle11\bar{2}6\rangle]$ has a comparable EM in the initial stage of the slip but the actual EM occurring at the slip distance of 6.5 Å is very high (65,280 mJ/m²). The pyramidal $c + a$ slip $[(11\bar{2}\bar{1})\langle2\bar{1}\bar{1}3\rangle]$ encounters a high EM (29,680 mJ/m²) immediately at the beginning of the slip. All the slip systems for which we calculated PES are shown in Supplementary Fig. 7.

The most important observation from PES is that for SmCo₅ even the lowest EM is one order of magnitude higher than the values calculated for hcp metals (Supplementary Table 4). In addition, the energy introduced into a crystal by a dislocation is proportional to $b^2$, where $b$ is the length of the Burgers vector[24]. The values of $b$ for dislocations in SmCo₅ are above 5 Å, significantly larger than, for instance, the value of 2.55 Å for Cu $[(111)\langle\bar{1}10\rangle]$ and 3.21 Å for Mg $[(0001)\langle11\bar{2}0\rangle]$ slip systems. One should note that dislocations can be observed in hard or complex materials even if they have a high EM on PES, but in that case they have a small value of $b$[26–30] or when deformation is confined in a very small volume[31]. For instance, partial dislocation has even been found the Cr₂Hf Laves phase[27], which has a high EM (2900 mJ/m²)[28] but a small value of $b$ (1.5 Å). In Laves phases the existence of partial dislocations relies on alternative stacking of lattice planes, which geometrically is not allowed in SmCo₅.

While no complete dislocation slip has been observed in our MD simulations, we have observed incomplete slips on pyramidal

$2c + a$ slip system, as shown in Fig. 2b. This incomplete slip can be understood by examining the PES in Fig. 2a, where the energy at the beginning of the slip is relatively low and the slip becomes arrested at the distance smaller than ~1.5 Å because it encounters a high EM (Fig. 2a). Incomplete pyramidal slips are illustrated in Fig. 2b, where atoms are color-coded by their displacements from the original position in an unstrained sample. Atomic-level view of the pyramidal slip is shown in Fig. 2c. The slip distance observed during simulations is significantly smaller than the length of the Burges vector, which is 9.5 Å. Although a number of the $2c + a$ slips can be observed in a single grain, the amount of strain that can be accommodated in this way is limited due to the incompleteness of the slip. The incomplete slip in SmCo₅ does not lead to formation of stacking faults, which have been found for instance in Laves phases[27]. Basal slip is not found either, which is likely due to the fact that the probability that the single basal plane will have the highest resolved shear stress is lower than the probability that one of the three planes for the $2c + a$ slip will have high enough resolved shear stress to initiate slip (Fig. 2c), even if basal slip has the lowest EM. It is obvious that SmCo₅ does not have five independent dislocation slip systems (actually it has none) that can continue accommodating strain to guarantee an arbitrary plastic deformation of a polycrystal. Normally, a material with these properties would be brittle[32]. However, this is not the case here since extensive plasticity has been clearly demonstrated by our MD simulation.

**Amorphous shear bands**. Our simulations reveal that plasticity in SmCo₅ is driven by GB sliding and by direct amorphization along non-crystallographic planes. We refer to the resulting structures as amorphous shear bands. Typically, in polycrystalline samples GB sliding requires so-called accommodation mechanisms, which are mechanisms that release stress accumulated at a triple junction due to sliding of an adjacent GB. We find that nucleation of thin amorphous shear bands provides an accommodation mechanism in polycrystalline SmCo₅. This process is illustrated in Fig. 3a–d. When a resolved shear stress on a GB exceeds the GB shear strength, the GB tends to slide (Fig. 3a) and there is accumulation of local stresses at the triple junction (Fig. 3b). The local stress resists further slip along the GB and it increases with strain. After the

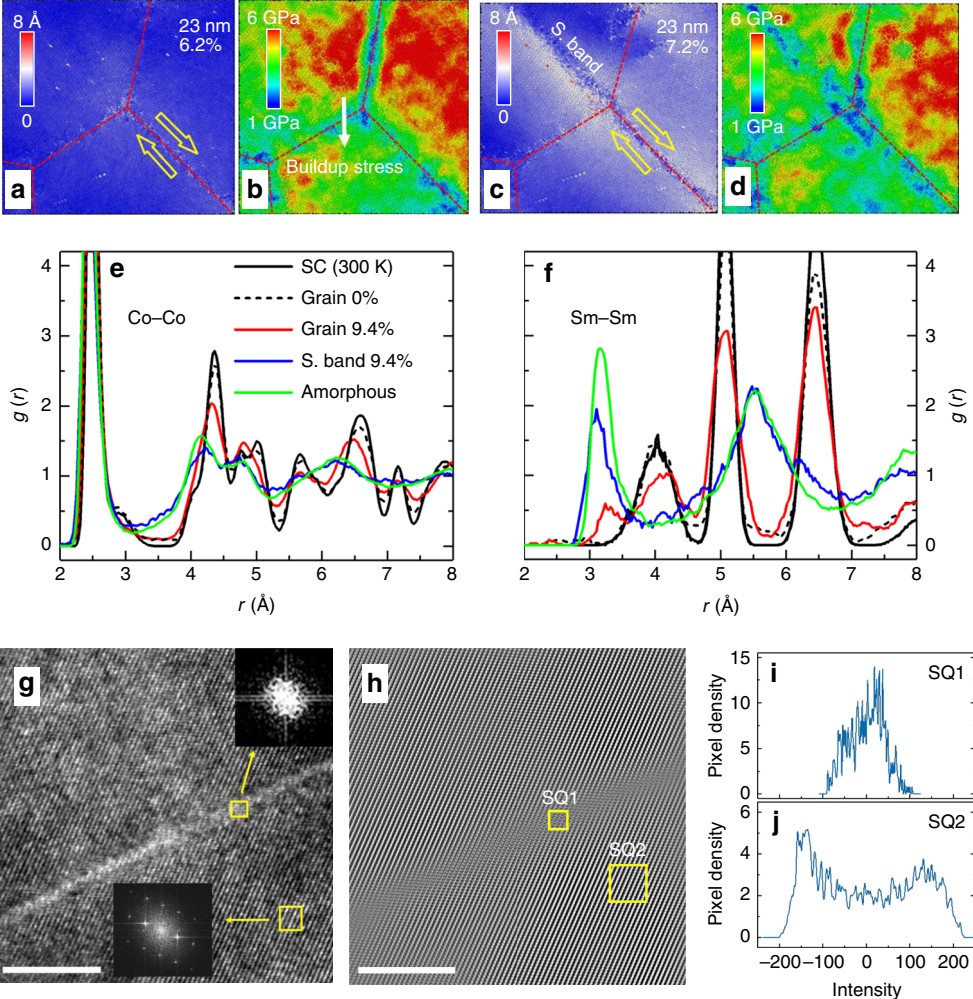

**Fig. 3** Amorphous shear bands. **a–d** Displacements of atoms relative to their positions in the unstrained samples and distribution of von Mises stress in the same area around a triple junction for grain size of 23 nm under the strain of 6.2% (**a**, **b**), and the strain of 7.2% (**c**, **d**), respectively. **e**, **f** Pair distribution functions [$g(r)$] of Sm–Sm (**e**) and Co–Co (**f**) for a bulk single crystal at 300 K (SC), unsheared crystalline region (Grain) in the grain with zero and 9.4% strain, shear band (S. band) under 9.4% strain, and the bulk amorphous state (quenched from 2000 K). **g** The HRTEM image of a selected shear band and its surrounding regions. FFT patterns are shown in the insets. Scale bar, 10 nm. **h–j** The inverse FFT images: **h** shows a blurred stripe corresponding to the amorphous shear band. Scale bar, 10 nm. The pixel distribution of intensity is unimodal (**i**) and bimodal (**j**) for the shear band and surrounding regions, respectively, indicating amorphous and crystalline characters of these regions[51]

stress has reached a critical value, it leads to nucleation of an amorphous shear band from the triple junction (Fig. 3c). The nucleation event releases local stress as shown in Fig. 3d, and the GB sliding is able to continue, accompanied by a sudden softening before the stress is stabilized. In addition to serving as accommodation mechanisms for GB sliding, the shear bands on non-crystallographic planes can be nucleated inside the grains and without being initiated by a GB sliding event (Supplementary Figs. 8 and 9). Orientations of the shear bands are at approximately 45° to the loading axis since this is the direction of the maximum resolved shear stress and they do not correspond to any specific orientations of the crystal. The same mechanisms are observed for both compressive and tensile deformation.

To demonstrate that the regions within the slip are indeed amorphous, we have calculated pair distribution functions (PDFs) of Co–Co and Sm–Sm in local regions of the grain and compared them with the corresponding PDFs in a single crystal (SC) and an amorphous bulk (quenched from the liquid state at 2000 K). The results are shown in Fig. 3e, f. The unsheared zone (grain) before deformation is clearly crystalline as it has PDFs very similar to those of SC. For samples strained in compression to 9.4%, the

peaks in the PDFs of the crystalline zone are broader and lower due to local distortions but still have a clear resemblance to those in SC. In contrast, PDFs calculated within the shear band match well with those of bulk amorphous SmCo$_5$ sample and no grain size dependence of the PDFs is found. It is important to mention that the EAM potential was explicitly fitted to the equations of state not only of the crystalline phases of Sm–Co, but also of amorphous phases and therefore it can capture the energetics of phase transformation in this alloy (see Supplementary Note 1). In addition, it is known from experiments that SmCo$_5$ may be a good glass former, as for instance it can be amorphized by high-energy ball-milling[33].

Note that the term shear band has been sometimes used in literature to refer to other unrelated processes in deformation. Specifically, in polycrystalline metals a shear band has been commonly used to describe a mesoscale region of localized dislocation activity and twinning[7,34]. Such shear bands are generally large (sometimes micrometer thick), and they contain band-like dislocation networks or twin lamellae. In contrast, here the term shear band refers to a region that forms in the process of direct amorphization along a highly stressed plane, and that does

not require the presence of dislocations. In our case, the amorphous shear band is the primary driver of plasticity and it is approximately 2 nm thick (without noticeable dependence of the thickness on the grain size). Amorphous shear bands similar to ours have been previously observed in shock-loaded covalent ceramics (i.e., $B_4C$ and SiC)[35,36]. However, they were initiated either along certain crystallographic planes ($B_4C$) or by planar faults (SiC) and, without exception, they provided the dominant mode of brittle fracture[37–39]. The term shear band has also been used in the context of metallic glasses to refer to thick (often about 20 nm) regions of localized deformation in a material that was already amorphous prior to mechanical loading. In metallic glasses shear bands are also the primary source of cracks[40]. In contrast, here, the amorphous shear bands are formed during a process of crystalline-to-amorphous transformation along non-crystallographic planes and they do not evolve into cracks but instead they enable plasticity in the absence of dislocations.

To verify our predictions that $SmCo_5$ can undergo large plastic deformation, we have deformed experimentally micropillar samples, as shown in Supplementary Fig. 10. We found that indeed $SmCo_5$ micropillars deform plastically and do not fracture when subjected to more than 20% of engineering strain. To test theoretical predictions of stress-induced amorphous shear bands, we also performed high resolution transmission electron microscopy (HRTEM) analysis of a $SmCo_5$ sample that was deformed by indentation (see Methods). We found multiple shear bands in the deformed region (Supplementary Figs. 11 and 12). In Fig. 3g, we show an example of an HRTEM image of a shear band and its surrounding regions. We do not find any dislocations in these regions. Fast Fourier transform (FFT) and inverse FFT patterns clearly show that the shear band is amorphous whereas the off-band regions are crystalline (Fig. 3h–j). It is also found that the amorphous shear bands propagate in the grains without initiation of cracks.

**The role of growth dislocations**. Although it is energetically unfavorable to nucleate dislocations in $SmCo_5$ during deformation, dislocations can be introduced into this material during growth, as reported from experiments in ref. [41]. The most common dislocations observed in these experiments lie in the basal plane with the Burgers vector $\langle 11\bar{2}0 \rangle/3$. The growth dislocations are not present in our samples since our samples were prepared using grains cut out from a perfect crystal. To test if growth dislocations can accommodate plastic strain during deformation, we introduce an artificial dislocation in the basal plane of an orthorhombic grain with the Burgers vector $\langle 11\bar{2}0 \rangle/3$. We then relax the atomic positions in MD simulations performed at 300 K, and we obtain the structure of a stable dislocation. This dislocation has a width of about 4 nm. Subsequently, we apply uniaxial compressive stress along the axis tilted at 45° with respect to the basal plane. Even though there is a pre-existing dislocation, the stress–strain curve (Supplementary Fig. 13a) shows a high yield strength (3.96 GPa), as compared to the highest yield strength of 3.88 GPa measured in the polycrystal sample with the grain diameter of 65 nm. We find that the dislocation does not move in a manner typically observed in metals under the shear stress. Instead, the dislocation acts as a precursor for amorphization (Supplementary Fig. 13b–e), which begins at the point of yield (corresponding to a strain of 7.8%). At the strain of 8.3% (corresponding to the strain immediately after yielding), a thin amorphous shear band extends along the plane with maximum resolved shear stress (which in this case is the same plane on which the dislocation lies) and stretches across the entire grain. Plastic deformation beyond this point proceeds in a manner similar to what we reported for the polycrystalline samples.

## Discussion

Having identified mechanisms of plasticity in $SmCo_5$, we can now interpret the grain size dependence of these mechanisms. For grain diameters below 15 nm, most of the strain is accommodated by the GB sliding, as shown in Fig. 4a. This is consistent with the simulated results for nanograined Cu[9] and Ni[13,42]. For larger grain sizes, we observe both GB sliding and formation of amorphous shear bands (Fig. 4b). As shown in Fig. 4c, the incomplete pyramidal $2c + a$ slip saturates at the strain of ~6.2% and has a maximum displacement of ~1.5 Å, irrespective of the grain size. In contrast, GB sliding distance increases with the grain size and it does not saturate easily because it is accommodated by formation of shear bands. These observations can be explained by the fact that materials with larger grains have a lower fraction of GBs, thus individual GBs have to slide a longer distance to accommodate the same external strain than in the small-grained samples (Supplementary Fig. 14). As a result, the back stress at individual triple junctions increases with increasing grain size, which explains why samples with larger grains have a higher strength. When the grain size is large enough, the local stress at the triple junctions may reach a critical value, activating amorphous shear bands. Since propagation of shear bands continues accommodating GB sliding, the back stress at triple junctions cannot exceed the critical value and increasing the grain size further does not result in the increase of materials strength. These processes lead to the extended inverse HP relation and plateauing of strength at larger grain size observed in our simulations, as schematically shown in Fig. 4d. It should be mentioned that although GB diffusion is important to the deformation of some nanograined materials[43,44], our additional nanoindentation tests for different strain rates (Supplementary Fig. 15) indicate that it may not be significant in $SmCo_5$. In general, it can be concluded that an inverse HP relation spanning over a wide range of grain size is attributed to the difficulty of nucleation and motion of dislocations, which otherwise may dominate the strain accommodation to enable dislocation plasticity and lead to a traditional HP relation.

Typically, lack of dislocation plasticity would result in a brittle failure of the sample, likely with cracks initiated at GBs, as has been observed in other nanocrystalline materials[21,45]. The ability of the $SmCo_5$ to form amorphous shear bands leads to suppression of fracture and to an increase of plasticity of this material. One should note that if there were large flaws introduced into $SmCo_5$, e.g., during powder sintering, these external flaws could lead to brittleness of $SmCo_5$. The effect of external flaws on deformation of $SmCo_5$ is not considered in the current study because it is strongly dependent on the synthesis conditions. It will be also interesting to determine how the results from $SmCo_5$ can be generalized to search for other intermetallic systems that exhibit amorphous shear bands during deformation. Two features that are likely to play important roles are a high-energy barrier to dislocation glide (perhaps related to the complexity of the unit cell and large Burgers vectors) and low-energy barrier to amorphization (perhaps related to good glass forming ability of the materials).

## Methods

**Calculation of PES**. The PESs are energy profiles corresponding to the slip of atoms within a crystalline plane. To calculate the PESs, a supercell was constructed based on a relaxed structure for each slip system with a vacuum slab of 10 Å. The six top atomic layers were shifted along the slip direction without incorporation of relaxation[46]. The energy of the initial structure was taken as the reference. The PES was evaluated by both DFT and EAM potential. We used the Vienna ab initio simulation package (VASP)[47] to perform the spin-polarized DFT calculation with PAW pseudopotentials and generalized gradient approximation exchange correlation functionals[48]. The $f$ electrons of Sm are not considered in the valence states as they are localized and do not have a significant effect on bonding that determines the PESs. The $k$-point grids were chosen separately for different slip systems (corresponding to different supercells) and were sufficiently dense based on our convergence tests. MD calculations of PESs were done by using the large-scale

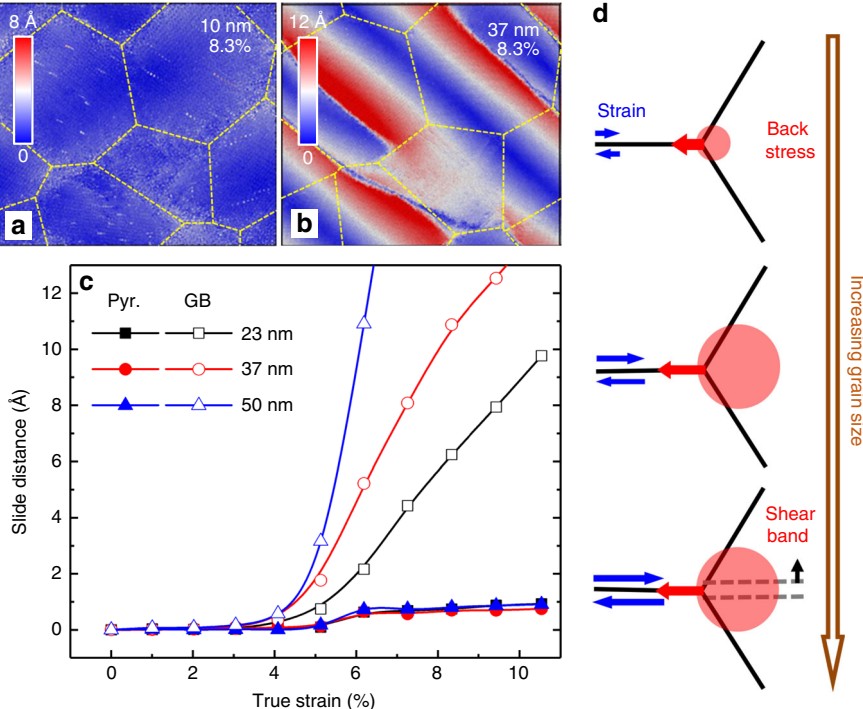

**Fig. 4** Grain size dependence of strain accommodation. **a, b** Displacements of atoms relative to their positions in the unstrained samples under the strain of 8.3% for grain sizes of 10 nm (**a**) and 37 nm (**b**), respectively. **c** Displacements as functions of applied strain for a pyramidal $2c + a$ slip (shown in Fig. 2) and the GB sliding (shown in Fig. 3) for samples with different grain sizes. The same location of slip/sliding was chosen in each sample. Examples from other locations show the same qualitative trend (Supplementary Fig. 14). **d** A schematic illustration of how GB sliding distance results in back stress for samples with different grain sizes

atomic/molecular massively parallel simulator (LAMMPS) code[49] with a newly developed EAM potential (see Supplementary Information).

**MD simulations of deformation**. Nanocrystalline samples containing ten grains with random orientations were prepared using Atomsk software using a cubic box with periodic boundary conditions[50]. The grain orientations are kept unchanged when grain size is varied from 5 to 65 nm. Before simulations, a small number of atoms at the GBs with distances shorter than 2 Å were removed due to their unphysically high energies (the closest atoms in $SmCo_5$ have a distance of 2.4 Å). First, we equilibrated the system at 20 K and zero pressure in NPT ensemble for 100 ps. The temperature was then gradually increased to 600 K, where the samples were annealed for 500 ps to enable sufficient atomic rearrangement and diffusion at the GBs. Subsequently, the samples were cooled gradually to room temperature where they were equilibrated for 250 ps. Uniaxial deformation along the z direction was simulated with a strain rate of $10^8$ s$^{-1}$ while the pressures in the x and y directions were kept zero. The largest engineering strain is 20% for compressive load and 30% for tensile load, corresponding to a true strain of 22.3% and 26.2%, respectively. The Nosé–Hoover thermostat was adopted in the NPT ensemble (zero pressure along the x- and y-directions) during all simulations. In simulations of deformation, we used the time step of 5 fs. For simulations of amorphization by melting (at 2000 K) and quenching the samples, the time step was set as 1 fs.

**Experimental sample preparation and indentation**. Bulk $SmCo_5$ magnets with tailored grain size were prepared by compacting powder precursors at temperatures of 500–700 °C followed by thermal annealing. To obtain bulk samples with grain sizes below 60 nm, amorphous precursors in $SmCo_5$ composition were firstly produced by ball-milling commercial $SmCo_5$ powder particles using a SPEX 8000D Mixer/Mill. The milling time was 3 h and the ball to powder ratio was around 15:1. The bulk sample with grain size about 7 nm was then produced by powder compaction and crystallization at 500 °C for 10 min under a pressure of 1.1–1.4 GPa. The bulk samples with grain sizes of 12, 21, 40, and 58 nm were then obtained by annealing the 500 °C as-compacted samples at 550, 700, 725, and 750 °C for 30 min, respectively. To prepare bulk samples with grain sizes of 78, 90, and 98 nm, crystalline precursors with milling time of 5 min and grain size about 15 nm were compacted at 700 °C for 10 min followed by respective annealing at 700 °C for 30 min, 725 °C for 30 min, and 750 °C for 60 min. Samples with grain sizes in the micrometer regime were prepared from crystalline precursors with grain diameter of about 3 μm by compaction and annealing the at 1150 °C for 1 h. When raising the temperature, the heating rate was 30–50 °C/min. After annealing, the samples

were cooled naturally to ambient temperature. The samples have cylindrical shapes with a diameter of 10 mm and a height of about 9 mm.

Archimedes method was used to examine the density, and the density of the bulk samples was determined to be about 8.4 g/cm$^3$, which is over 98% of the full density. The crystal structure of the bulk magnets was identified by X-ray diffraction (XRD) using Cu-$K_a$ radiation and the grain size was estimated by Scherrer's formula, as shown in Supplementary Fig. 6. XRD pattern shows the dominance of the $SmCo_5$ phase and there is no clear signature of other phases in the samples. For micro Vickers hardness measurement, a condition of a 200–300 gf load and a 10 s dwell time was used. Each bulk sample was tested more than 10 times to get the average hardness value.

Nanoindentation was performed by Hysitron TI 950 TriboIndenter equipped with a diamond Berkovich tip. The sample with grain diameter of 12 nm was first polished until the surface root mean square roughness is less than 3 nm in the area of 3 μm × 3 μm. The effects of loading rate and dwelling step were measured as follows: (i) the loading rate was set from 100 to 3000 μN/s with a step of 100 μN/s. Ten nanoindents were measured for each loading rate. The maximum loading was set as 6000 μN with the dwelling time of 10 s and the size of the nanoindentation of ~700 nm; (ii) dwelling steps with 10, 20, 50, 100, 200, 500, and 1000 s were used. Ten nanoindents were measured for each dwelling step. The loading rate was set as 500 μN/s and the maximum loading was 6000 μN.

**Deformation of micropillars**. The micropillar samples were prepared using the sintered sample with micrometer-sized grains by a FEI Helios PFIB G4 FIB/FESEM Focused Ion Beam (FIB) instrument in the Materials Science Center at the University of Wisconsin-Madison. The micropillars have nominal lengths ranging from 7 to 9 μm and have diameters of 1.5 μm on the top and 3 μm at the bottom. The micro-compression experiments were performed in Zeiss Leo 1550VP SEM equipped with a Hysitron PI85 SEM Pico indenter using a diamond-made flat-ended conical nanomechanical probe. The nominal flat diameter is 5.1 μm and the nominal cone angle is 60°. The maximum displacement of the indenter was 2000 nm, finished in 5 s with a uniform speed.

**TEM observation of deformed samples**. Samples for TEM analysis were prepared using standard lift-out techniques with FIB system (see Supplementary Fig. 11). To protect the sample surface from damage during FIB preparation, a 4.0 μm Pt protective layer was deposited on the surface of the indented region by two steps: (i) 2 kV electron beam (low energy) was used to deposit a 1.0 μm Pt layer to avoid damage from high-energy ions deposition; (ii) a 12 kV ion beam was used for the deposition of another 3.0 μm Pt layer. The thinning process was speeded up by

high-energy ion beam (30 kV) at the beginning and ended with a low-energy ion beam (2 kV) to carefully remove the amorphous area generated in the former stage. FEI Tecnai F30 with field emission gun TEM and high resolution TEM were used to identify amorphous shear bands.

## Data availability

The data sets generated during and/or analyzed during the current study are available from the corresponding author on reasonable request.

## Code availability

Simulations have been performed using publicly available software packages (LAMMPS, VASP, OVITO).

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

## Acknowledgements

H.B.L. acknowledges the financial support from the program of China Scholarships Council (No. 201704910234) and the continuous support of computing resource from CCMS of the IMR in Tohoku University. I.S. and H.Z. acknowledge support from ARO grant No. W911NF-17-1-0571. J.D. acknowledges support from the NSFC grant (No. 51771220). F.Q.W. acknowledges support from Baotou Rare Earth Research and Development Center (No. GZR2018002).

## Author contributions

I.S., H.B.L., and J.P.L. conceived the study. H.B.L. carried out all the simulations. H.B.L. and I.S. analyzed the data and co-wrote the paper. H.W.S. developed potential for MD

simulations and contributed the corresponding sections in the paper. H.L.Z carried out all the micropillar deformation, nanoindentation, and TEM experiments. H.W.S. and J.P.L. provided comments on the study and the paper. F.Q.W., J.K.F., and J.D. synthesized the experimental samples and carried out measurements of Vickers hardness.

## Additional information

**Competing interests:** The authors declare no competing interests.

