## [Peer Review File · Nature Communications]

Reviewers' comments:

Reviewer #1 (Remarks to the Author):

The manuscript entitled "Plasticity without Dislocations in a Polycrystalline Metal" is mainly focused on the theoretical insights of deformation mechanisms of an intermetallic SmCo₅ compound. Grain size dependent plasticity is investigated on this system and successfully predicted inverse Hall-Petch relation in a non-cubic crystal system. This study opens up the investigations of extended plasticity in hexagonal intermetallic compounds. However, I have major concerns with the current manuscript as comments mentioned below.

1. The work mainly focused on the SmCo₅ which is an intermetallic compound. However, the title contains "metal" which needs to be changed to "intermetallic" or specific to "SmCo₅ compound". In fact, throughout the manuscript the word metal and metallic need to be corrected at appropriate places with intermetallic. (Typo error – (In line 106) it must be SmCo₅ rather than 'SmCo₅'.)

2. It is stated that SmCo₅ is a brittle material as it does not have any slip systems (line 112). It means the material SmCo₅ (having grain size in micron range) shows brittle nature. Please cite references to support this statement (either theoretical or experimentally). If not, experiments or simulations need to be performed to understand the brittle nature in micro-size grains.

3. The authors claimed that the material SmCo₅ can undergo large plastic deformation (18%) through the simulations. However, there is no experimental evidence on such a huge plasticity in the manuscript. The hardness (H) obtained from indentation technique is a localized deformation and can relate to the strength (σ) of material only, which is $\sim (\sigma=H/3)$. It would be better to perform micro-tensile or compression tests which are more reliable and accurate in order to understand the real deformation behaviour of SmCo₅.

4. In reported paper (T. Saito et al., Science (2003) 300, 464-467) also shows dislocation-free plasticity in metals (Gum metals). They reported that there are some magic numbers (Bond order (Bo), d-electron orbital energy level (Md), and electron-to-atom (e/a) ratio) has to satisfy with certain number (Bo = 2.87 : Md = 2.45 eV : e/a = 4.24) simultaneously to occur such deformation mechanism apart from the experimental conditions. This kind of intimate numbers is very useful in order to screen the alloys rapidly.

It would be very helpful if authors of this paper could show some theoretical insights to identify such systems. It is interesting since so far inverse Hall-Petch relation is valid for mainly fcc systems, whereas now it is extending to other crystal systems.

5. Authors need to provide experimental details on sample preparation. Sintering processing technique used to achieve bulk samples? What are heating/cooling rate? What is size of the sample that you prepared?

6. Line 36-38, authors have stated that only traditional HP (Hall-Petch) relation has been reported for HCP systems. However, there are some reports on the HIP (Inverse Hall-Petch) on HCP (Hexagonal Close Packed) alloys such as Mg and Zr (H.Y. Song et al., Journal of Applied Physics (2012) 111, 044322 and C.J. Ruestes et al., Scripta Materialia (2014) 71, 9-12). Please don't make such bold statements without screening the existing literature. Hence, the statement has to be modified according to the previously reported results.

7. Also, it was found from literature (Reddy et al. Nature communications (2012) 3, 1052 and D Guo et al. Physical Review Lett. (2018) 121, 145504) in a superhard nanocrystalline boron carbide, the GB sliding which was associated with amorphization.

Similarly, authors have made observations on plasticity of SmCo₅ by MD simulations and explained the fundamental deformation mechanism process (Line 117-135). So, I suggest the authors to cite these two existing literature works.

That said, the observations found by authors is very interesting on SmCo₅ intermetallic system, certainly suitable for Nature communications, and I would advise to consider a revised manuscript, provided that the above mentioned questions will be addressed by the authors.

Reviewer #2 (Remarks to the Author):

The manuscript submitted by Luo et al. deals with the deformation of nanocrystalline SmCo₅, a material used in magnets. Luo et al. describe their computational efforts that, surprisingly, show deformation by formation of amorphous bands, rather than dislocations, and the interaction of these bands with grain boundaries. The authors compare their results with experimental evidence they also collected and rationalise their observations by combining the two.

I enjoyed reading this manuscript and believe it is of interest to the broader scientific readership of Nature Communications as well as of novelty and interest to those working in crystal plasticity in particular.

The manuscript is well written and the images prepared with great care. A few comments and questions are listed below, which I would ask to address prior to acceptance/publication. You will no doubt be able to tell that I am an experimentalist – please feel free to answer potentially pedestrian comments with respect to the computational work accordingly.

- From my perspective, the main novelty described here (and indeed featuring prominently in the title) lies in the absence of dislocations. While the introduction is well-written and logical, it might be good to consider restructuring it slightly to start with the missing defect rather than a general explanation of the Hall-Petch effect.

- On page 9, the authors comments on the suppression of fracture in SmCo₅ and the consequently increased plasticity. I cannot see how this is arrived at: the stress-strain curves in the Supplemental Information show strain softening, which in technical applications/testing at increasing load leads to sudden failure. Experimentally, the authors refer to indentation testing. This method is commonly used to suppress cracking in even the most brittle materials, such as other intermetallics and glasses. Indeed, the TEM micrograph in the experimental section shows a crack. I would therefore recommend removing or limiting this conclusion to certain conditions or supporting it by further evidence.

- With reference to the mentioned conventional TEM image: As this is provided, I would feel readers received a much better ability to check the obtained results and compare with their background if a larger part of the plastic zone was shown or a SAD pattern was included to again compare the orientation of the shear bands with potential crystal planes.

- Minor comment on style in Fig S9: Please remove the platinum pattern still captured as part of the indentation and refer to the indentation simply as that, it is not a hole, just an indentation or imprint.

- The authors comment on the high energies associated with shearing of the SmCo₅ crystals. In this context they mainly refer to metals for comparison. However, as a highly ordered intermetallic, I would believe other intermetallics or further hard crystals may be of better use for comparison. An obvious one, because it has been studied so widely as a hard extreme, may be silicon. In fact, silicon has a similar USF energy as reported on the basal plane here of nearly 2 J/m² and Burgers vector on the shuffle set of 3.84 Å (Kaxiras E, Duesbery MS. Free energies of generalized stacking faults in Si and implications for the brittle-ductile transition. Physical review letters. 1993 Jun 14; 70(24):3752.). Please comment on this or expand the comparison with the literature. In silicon, these dislocations are in fact observed experimentally and in simulations, so that the values themselves cannot explain the findings here. Similar values, of the order of 1 J/m² are found in Laves phases.

- In this context it might in fact be equally valuable to highlight the opposite, namely that in many complex intermetallics crystals dislocations are in fact observed, even in quasi-crystal approximants (Nature materials. 2010 Apr; 9(4):332 & Scripta Materialia, 2018. 146: p. 327-30).

- The comparison of grown-in and mobile dislocations performed by the authors is a very valuable one. For basal slip the authors find that the reported grown-in dislocations are not mobile. As an experimentalist looking at the orthorhombic “grain”. I am slightly worried about the boundary conditions of this simulation. Can simple shear occur, i.e. is this effectively a micropillar or would shearing across the grain introduce a stress in periodic images of the “grain” that may impede crystallographic slip more effectively than amorphisation?

- Fig 1: Similar to above, the authors compare their work with one of the softest available metals. A better comparison may at least be cobalt itself. Nanocrystalline cobalt was found to possess a tensile strength of 2 GPa (A.A. Karimpoor, U. Erb, K.T. Aust, G. Palumbo, High strength nanocrystalline cobalt with high tensile ductility, Scripta Materialia, Volume 49, Issue 7, 2003, Pages 651-656) , of the same order as the commonly assumed third of the Vickers hardness reported here of (7/3 GPa).

- HR-TEM: The authors should comment on how they can be sure that crystallinity is in fact absent from their micrographs. The HR-TEM micrograph in the Supp Info appears to still show partial crystallinity. How would the observed effect differ if slip was out of plane such that the image within the band was no longer "on axis"?

Reviewer #3 (Remarks to the Author):

In this paper, the authors reported a comprehensive investigation on the grain-size dependent deformation behavior in SmCo₅, an intermetallic with a space group P6/mmm (hexagonal but not close-packed). The primary findings are the (a) absence of traditional dislocation accommodated Hall-Petch hardening in the coarse-grain regime due to the shear band plasticity, and (b) GB softening in the smaller grain size regime due to grain boundary sliding. The paper is well organized and the conclusion is convincing. The novelty of the report should be strengthened for for competition of publication in Nat Comm. Primary reasons are given below:

(1) It has been broadly accepted that dislocation sources and pile up are hardly expected to exist in nc materials because of an increasing percentage of grain boundary atoms and a grain boundary accommodation mechanism should be responsible for the plastic deformation. With increasing grain boundary volume, the soft region, macroscopic strength softening is expected. This phenomenon seems to be independent of metallic structures. Hence the softening observed in SmCo₅ is not surprising.

(2) From the references of the paper, deformation induced shear localization, which becomes the primary plasticity carrier, has been documented in literature, see ref. 24,25,26, and 34. It would be important to demonstrate atomic details of the amorphization process as a resultant of shear localization.

(3) It is important to show whether such shear bands exhibit grain size dependent behavior, e.g., structure, width, ... That could make the paper stand out from existing research reports. There are further minor issues to be addressed.

In presenting the mechanical behavior of SmCo₅ as a function of grain size and made a comparison with Cu, the deformation mode for both simulations and experiments, tension/compression, constant volume/constant pressure..., should be specified clearly.

Many of strongly related references are missing. For instance, in both BCC and HCP metals, both traditional HP relation and the inverse one were reported.

grain boundary deformation has been documented in many pioneer work, from MD, Experiments, FEM simulations and so on:

Schitz, J., Di Tolla, F.D. & Jacobsen, K.W. Softening of nanocrystalline metals at very small grain sizes. Nature 391, 561–563 (1998).

Van Swygenhoven, H., Spaczer, M., Caro, A. & Farkas, Competing plastic deformation mechanisms in nanophase metals. Phys. Rev. B 60, 22–25 (1999).

Shan, Z.W. et al. Grain boundary-mediated plasticity in nanocrystalline nickel. Science 305, 654–657 (2004).

Wei, Anand. Grain-boundary sliding and separation in polycrystalline metals: application to nanocrystalline f.c.c. metals, Journal of Mechanics and Physics of Solids, 52, 2587-2616 (2004).

In addition, is grain boundary diffusion an important mechanism for plasticity in SmCo₅ at sufficiently small grains? Such mechanism seems to be important for FCC metals, see Gao et al. Enhanced strain-rate sensitivity in f.c.c. nanocrystals due to grain-boundary diffusion and sliding,

Acta Materialia, 56, 2008, 1741-1752. A possible way to examine is to using the indentation tests at different loading rate to check the rate-sensitivity and also using dwelling step to see stress relaxation.

Reviewers' comments:

Reviewer #1 (Remarks to the Author):

1. The work mainly focused on the SmCo₅ which is an intermetallic compound. However, the title contains “metal” which need to be changed to “intermetallic” or specific to “SmCo₅ compound”. In fact, throughout manuscript the word metal and metallic need to correct at appropriate places with intermetallic. (Typo error – (In line 106) it must be SmCo₅ rather than ‘SmoCo₅’.)

We agree with reviewer that “intermetallic” is more specific than “metal”. We have modified the manuscript accordingly. The typo error has also been corrected. Thank you.

2. It is stated that SmCo₅ is a brittle material as it does not have any slip systems (line 112). It means the material SmCo₅(having grain size in micron range) shows brittle nature. Please cite references to support this statement (either theoretical or experimentally). If not, experiments or simulations need to perform to understand the brittle nature in micro-size grains.

There seems to be a misunderstanding of what we meant to say in line 112 (now 126). We state that normally if the material did not have five independent dislocation slip systems, it would be brittle. This is provided that no other deformation mechanism is operative. SmCo₅ is not inherently brittle (despite not having active dislocation slip systems) because of the newly discovered mechanism of amorphization shear bands. Per reviewer’s suggestion we have added a reference to support the statement that materials become brittle in the absence of independent slip systems (Ref. 33). We have also modified the phrase “five independent slip systems” to “five independent dislocation slip systems” so that the slip systems will not be confused with shear bands found in this study.

As a separate note, the mechanisms found in our simulations are not limited to small grain sizes. In fact, the experimental evidence we provided for plasticity and existence of amorphous shear bands was collected on samples with micrometer-sized grains.

3. The authors claimed that the material SmCo₅ can undergo large plastic deformation (18%) through the simulations. However, there is no experimental evidence on such a huge plasticity in the manuscript. The hardness (H) obtained from indentation technique is a localized deformation and can relate to the strength (σ) of material only, which is $\sim (\sigma=H/3)$. It would be better to perform micro-tensile or compression tests which are more reliable and accurate in order to understand the real deformation behaviour of SmCo₅.

Thank you very much for the suggestion. We have performed micro-compression experiments [using a technique similar to that in *Nature Communication* 3, 205 (2012)] on SmCo₅ and we confirmed that the samples can be deformed plastically to more than 20% of engineering strain without fracture. The results are shown in a newly added Fig. S9.

4. In reported paper (T. Saito et al., Science (2003) 300, 464-467) also shows dislocation-free plasticity in metals (Gum metals). They reported that there are some magic numbers (Bond order (Bo), d-electron orbital energy level (Md), and electron-to-atom(e/a) ratio) has to satisfy with certain number (Bo = 2.87 : Md = 2.45 eV : e/a = 4.24) simultaneously to occur such deformation mechanism apart from the experimental conditions. This kind of intimate numbers is very useful in order to screen the alloys rapidly.

It would be very helpful if authors of this paper could show some theoretical insights to identify such systems. It is interesting since so far inverse Hall-Petch relation is valid for mainly fcc systems, whereas now it is extending to other crystal systems.

First of all, we would like to say that plasticity of gum metals is based on an entirely different mechanism. Gum metals exhibit superelasticity during deformation, which is usually accounted for by stress-induced reversible martensitic transformation [Acta Mater. (2009);57:1188]. As the reviewer mentioned, oftentimes occurrence of martensitic transformation can be correlated with features of the electronic structure of the material. For example, the e/a criterion is based on the Hume-Rothery electron concentration rule: Varying the alloy composition changes the number of electrons (thus changes e/a); when the Fermi sphere expands (if the number of electrons increases) and touches the edge of the Brillouin zone, the lattice tends to be destabilized [Crystals (2017);7:9]. We do not expect that a similar rule would apply to predict formation of amorphous shear bands instead of dislocations.

We agree that it would be useful to provide an insight about conditions for when an extended inverse HP relation will be manifested. However, to provide such insight from atomic or electronic perspective is still challenging and will be an interesting topic for future studies. Here, a conclusion that can be reached based on the current study is: In general, it can be concluded that an inverse HP relation spanning over a wide range of grain size is attributed to the difficulty of the nucleation and motion of dislocations, which otherwise may dominate the strain accommodation to enable dislocation plasticity and lead to a traditional HP relation.

5. Authors need to provide experimental details on sample preparation. Sintering processing technique used to achieve bulk samples? What are heating/cooling rate? What is size of the sample that you prepared?

We agree with this comment and we have added additional description of the sample preparation to the revised Supplementary Materials. Specifically, the bulk samples were sintered from powder precursors obtained by ball milling. For the sample with grain size of about 7 nm, the powder (partially amorphized) was heated to 500 °C and was compacted under a pressure between 1.1-1.4 GPa for 10 min. Samples with grain sizes of 12, 21, 40 and 58 nm, were obtained by further annealing the as-compacted samples at 550,

700, 725 and 750 °C, respectively, for 30 min. Samples with grain sizes of 78, 90, and 98 nm were prepared by compacting crystalline powders with grain size of 15 nm at 700 °C for 10 min followed by annealing at 700 °C for 30 min, 725 °C for 30 min and 750 °C for 60 min, respectively. For the sample with grain sizes in the micrometer regime, we used crystalline precursors with a grain size of about 3 μm and we compacted and annealed the samples at 1150 °C for 1 h. When raising the temperature, the heating rate was 30-50 °C/min. After annealing, the samples were cooled naturally to ambient temperature. Each sample has a cylindrical shape with a diameter of 10 mm and a height of about 9 mm.

6. Line 36-38, authors have stated that only traditional HP (Hall-Petch) relation has been reported for HCP systems. However, there are some reports on the HIP (Inverse Hall-Petch) on HCP (Hexagonal Close Packed) alloys such as Mg and Zr(H.Y. Song et al., Journal of Applied Physics (2012)111, 044322 and C.J. Ruestes et al., Scripta Materialia (2014) 71, 9-12). Please don't make such bold statements without screening the existing literature. Hence, the statement has to be modified according the previously reported results.

Thank you for pointing that out. We have modified the revised manuscript (lines 35-42) with this information. One should note that two of the references provided by the reviewer that show inverse HP relation are from MD simulations. We have found additional published MD simulations that show a similar behavior and included the references in the revised manuscript. A possible reason that generally inverse HP has not been observed in experiments on hcp metals is that the grain size in these experiments was not small enough.

Either way, published reports generally agree that for grain size larger than ~10-20 nm, there is typically a regime of traditional grain-size strengthening (consistent with HP relation) which regime is absent in our samples (both in simulations and experiments) because of lack of dislocation plasticity. Instead, SmCo₅ shows an extended regime of inverse HP followed by a regime where strength is insensitive to the grain size.

7. Also, it was found from literature (Reddy et al. Nature communications (2012) 3, 1052 and D Guo et al. Physical Review Lett. (2018)121, 145504) in a superhard nanocrystalline boron carbide, the GB sliding which was associated with amorphization.

Similarly, authors have made observations on plasticity of SmCo₅ by MD simulations and explained the fundamental deformation mechanism process (Line 117-135). So, I suggest the authors to cite this two existing literature works.

The two references are interesting with respect to the GB sliding and amorphization in nanocrystalline ceramics (boron carbide). Although the observed amorphization was found to be responsible for the brittleness of boron carbide, Reddy et al. showed that ductility can be improved by introducing nanoporosities. In the second reference, the lack

of dislocation activities in boron carbide also results in an inverse HP relation in nanocrystalline boron carbide as GB sliding dominates the strain accommodation before cracking due to amorphization. This is consistent with the general understanding that for very small grain sizes, GB sliding can dominate deformation and lead to inverse HP relation. We have cited these references (Refs. 39 and 40) in our revised manuscript.

That said, the observations found by authors is very interesting on SmCo5 intermetallic system, certainly suitable for Nature communications, and I would advise to consider a revised manuscript, provided that the above mentioned questions will be addressed by the authors.

We appreciate the valuable comments and suggestions from the reviewer.

Reviewer #2 (Remarks to the Author):

The manuscript submitted by Luo et al. deals with the deformation of nanocrystalline SmCo5, a material used in magnets. Luo et al. describe their computational efforts that, surprisingly, show deformation by formation of amorphous bands, rather than dislocations, and the interaction of these bands with grain boundaries. The authors compare their results with experimental evidence they also collected and rationalise their observations by combining the two.

I enjoyed reading this manuscript and believe it is of interest to the broader scientific readership of Nature Communications as well as of novelty and interest to those working in crystal plasticity in particular.

The manuscript is well written and the images prepared with great care. A few comments and questions are listed below, which I would ask to address prior to acceptance/publication. You will no doubt be able to tell that I am an experimentalist – please feel free to answer potentially pedestrian comments with respect to the computational work accordingly.

We are grateful to the reviewer for the positive comments and suggestions on how to improve the manuscript further.

1. From my perspective, the main novelty described here (and indeed featuring prominently in the title) lies in the absence of dislocations. While the introduction is well-written and logical, it might be good to consider restructuring it slightly to start with the missing defect rather than a general explanation of the Hall-Petch effect.

Thank you very much for thinking through the story line and for making suggestions on how to potentially improve it. There are really two surprising findings in our study. One is a new mechanism of plasticity in the absence of dislocation and the other

is lack of the traditional Hall Petch regime (observed in most metallic systems). These two observations are related to each other and the reviewer is correct that the manuscript could be written starting from either perspective. We have in fact written different versions of the introduction to see what makes for the best story. At the end of the day while we see the benefits of starting from the perspective of a missing defect, we feel that starting from the surprising grain size dependence, then asking the question about the deformation mechanism and finally revealing the surprising mechanism flows better and can more effectively capture the reader's attention. That said, we appreciate learning of the reviewer's reaction to the introduction and in response we modified the text to shorten the discussion of the background on the inverse Hall-Petch relation (which is perhaps not as surprising as the lack of traditional Hall-Petch behavior) and we fine-tuned the introduction to emphasize the open questions.

2. On page 9, the authors comments on the suppression of fracture in SmCo5 and the consequently increased plasticity. I cannot see how this is arrived at: the stress-strain curves in the Supplemental Information show strain softening, which in technical applications/testing at increasing load leads to sudden failure. Experimentally, the authors refer to indentation testing. This method is commonly used to suppress cracking in even the most brittle materials, such as other intermetallics and glasses. Indeed, the TEM micrograph in the experimental section shows a crack. I would therefore recommend removing or limiting this conclusion to certain conditions or supporting it by further evidence.

Again, these are very helpful comments. First, regarding the strain softening. The stress-strain curves the reviewer is referring to were generated from MD simulations and it is normal to see a higher yield stress and lower flow stress in MD simulations as long as the deformation mechanism has some activation energy barrier. One reason is that MD samples do not have pre-existing defects and dislocations/shear bands need to be nucleated (this difference between yield and flow stress is even more pronounced for simulated single crystal samples). Energy barrier to nucleating a defect is often larger than the energy barrier to move it. Secondly, MD simulations are obviously carried out at strain rates higher than those used in experiments and therefore the stress reached before defect nucleation (yield stress) can be quite high. Finally, in general static friction (e.g., during GB sliding) is typically higher than kinetic friction and this difference is further exacerbated in MD simulations because of higher strain rates. For these reasons one should be careful interpreting the evolution of flow stress with strain in MD simulations.

Given the strain-rate effects in MD simulations and potential issues discussed above, it is very reasonable to ask if the suppression of fracture and increased ductility would be present in experiments. The reviewer is correct that indentation can suppress fracture. Therefore (and following suggestions of the first reviewer), we have carried out additional experiments of micro-compression. These results are now included in Fig. S9 in the revised

manuscript. These experiments confirmed predictions from MD simulations that SmCo₅ can deform plastically without fracture to engineering strains larger than 20%.

Finally, the reviewer is correct that TEM image in Fig. S10d shows a needle-shaped crack. The reason may be traced back to sample preparation. A coarse-grained sample was sintered using crystalline precursors ball-milled from bulk SmCo₅ (obtained by smelting). The samples likely have imperfections (such as microvoids) that were not removed by sintering. These imperfections may initiate microcracks under loading. Regardless, our conclusion that amorphous shear bands in SmCo₅ can accommodate large plastic strain still applies to this case. It is true that if large pre-existing cracks were present in the sample, the ductility provided by amorphous shear bands may be insufficient to suppress fracture. Hence, qualifying the statement about suppression of cracking by amorphous shear bands in the experiment is well advised. We have modified the discussions accordingly in the manuscript (lines 191-192).

3. With reference to the mentioned conventional TEM image: As this is provided, I would feel readers received a much better ability to check the obtained results and compare with their background if a larger part of the plastic zone was shown or a SAD pattern was included to again compare the orientation of the shear bands with potential crystal planes.

Thank you for the suggestion. We have shown the indexed selected area diffraction (SAD) pattern of the background in the TEM graph (Fig. S10d). Please note that the zone axis is not the same as that in the HRTEM. We have also indexed the crystal planes in Fig. S11a and S11b for comparison. In order to find a lattice plane possibly parallel to the shear band, we would need to identify a vector in the reciprocal space that is indisputably vertical to the shear band (which is required if shear bands are lying in the lattice plane). This is a difficult task using the current technique. More importantly, the shear bands in the HRTEM images are not strictly out of plane (not perpendicular to the image plane) as they overlap with the lattice and there are still lattice fringes seen in the shear bands. We will address the issue of overlap in the response to comment 9 from the same reviewer.

4. Minor comment on style in Fig S9: Please remove the platinum pattern still captured as part of the indentation and refer to the indentation simply as that, it is not a hole, just an indentation or imprint.

We have modified Fig. S10 (previously Fig. S9).

5. The authors comment on the high energies associated with shearing of the SmCo₅ crystals. In this context they mainly refer to metals for comparison. However, as a highly ordered intermetallic, I would believe other intermetallics or further hard crystals may be of better use for comparison. An obvious one, because it has been studied so widely as a

hard extreme, may be silicon. In fact, silicon has a similar USF energy as reported on the basal plane here of nearly 2 J/m² and Burgers vector on the shuffle set of 3.84 Å (Kaxiras E, Duesbery MS. Free energies of generalized stacking faults in Si and implications for the brittle-ductile transition. Physical review letters. 1993 Jun 14;70(24):3752.). Please comment on this or expand the comparison with the literature. In silicon, these dislocations are in fact observed experimentally and in simulations, so that the values themselves cannot explain the findings here. Similar values, of the order of 1 J/m² are found in Laves phases.

6. In this context it might in fact be equally valuable to highlight the opposite, namely that in many complex intermetallics crystals dislocations are in fact observed, even in quasi-crystal approximants (Nature materials. 2010 Apr;9(4):332 & Scripta Materialia, 2018. 146: p. 327-30).

We agree with the reviewer that further comparing our results with those found in intermetallics or hard materials would make the discussion more complete. Apart from the energy maximum in the potential energy surface [comparable but strictly not always the same as the unstable stacking fault energy (USF) since the USF is defined as the lowest energy barrier for dislocation nucleation], the value of Burgers vector b is also important to the nucleation of dislocation as the elastic energy introduced into the crystal due to dislocation is proportional to b^2 . Thus, dislocations with larger values of b will be more difficult to nucleate and if formed, they tend to dissociate into partial dislocations.

As the reviewer mentioned, dislocations were found in other materials with high USF energies, such as Si (1.84 J/m²) and Cr₂Hf (2.9 J/m²). However, the reported dislocations in these materials are partials and have relatively short Burgers vectors. They are, 3.84 Å and 1.5 Å for Si and Cr₂Hf, respectively, as compared with the 5.04 Å for SmCo₅ in the basal plane. Similarly, in complex materials such as AlMnPd and Al₁₃Co₄ suggested by the reviewer, metadislocations in both cases were found to have a Burgers vector of about 1.8 Å in length. The glide of partial dislocations or metadislocations changes the stacking of the atoms (planar defects). In contrast, there is no alternative stacking in SmCo₅ (it is geometrically not allowed) that would enable partial dislocations analogous to those in AlMnPd and Al₁₃Co₄.

We have modified the discussion of dislocations to start by the fact that we do not observe any perfect or partial dislocations in our simulations. Lack of dislocations is then confirmed in experiments. We state that lack of dislocations is consistent with large EM on potential energy landscape and large Burgers vectors. We have also included discussion of other materials, as recommended by the reviewer. Finally, we referred to the USF energies adapted from references as energy maxima in our manuscript since the EMs are not always in accord with the definition of USF energy in our case.

7. The comparison of grown-in and mobile dislocations performed by the authors is a very valuable one. For basal slip the authors find that the reported grown-in dislocations are

not mobile. As an experimentalist looking at the orthorhombic “grain”. I am slightly worried about the boundary conditions of this simulation. Can simple shear occur, i.e. is this effectively a micropillar or would shearing across the grain introduce a stress in periodic images of the “grain” that may impede crystallographic slip more effectively than amorphisation?

To confirm that boundary conditions do not affect our conclusions, we have indeed carried out additional MD simulations with free boundary condition in the lateral directions, as shown in the figures below (colored by centrosymmetry parameter). The behavior of the grown-in dislocation is similar to what we showed in Fig. S12, i.e., dislocation serves as a precursor for amorphization under shear stress instead of gliding across the crystal. The reason specific boundary conditions do not play a role in our simulations is that the grain is sufficiently large and therefore the boundary effect on the dislocation is very small. We have added the discussion of the boundary effect in the revised caption of Fig. S12.

8. Fig 1: Similar to above, the authors compare their work with one of the softest available metals. A better comparison may at least be cobalt itself. Nanocrystalline cobalt was found to possess a tensile strength of 2 GPa (A.A. Karimpoor, U. Erb, K.T. Aust, G. Palumbo, High strength nanocrystalline cobalt with high tensile ductility, Scripta Materialia, Volume 49, Issue 7, 2003, Pages 651-656), of the same order as the commonly assumed third of the Vickers hardness reported here of (7/3 GPa).

The reason we have included data for nc-Cu in Fig. 1 is to show a typical transition from HP to inverse HP relation (Cu is a model material for which grain size dependence of strength was investigated both experimentally and in simulations extensively). In our discussion we do not compare the strength of SmCo₅ to the strength of Cu and we agree that such comparison would not be most interesting. We appreciate the suggestion of comparison to nc-Co. It is indeed an interesting observation that the strength of nc-Co for grain size of 12 nm is comparable to the strength of SmCo₅ with the same grain size. We have included this comment and a reference in the revised manuscript.

9. HR-TEM: The authors should comment on how they can be sure that crystallinity is in fact absent from their micrographs. The HR-TEM micrograph in the Supp Info appears to still show partial crystallinity. How would the observed effect differ if slip was out of plane such that the image within the band was no longer “on axis”?

It is true that there are still some lattice fringes shown in the HRTEM image of the amorphous shear band. These fringes do not result from crystallinity but from the overlap of the amorphous shear band with the lattice (the shear band is not perpendicular to the plane of the image). We have confirmed the amorphization by slightly adjusting the focus step and finding that the lattice fringes in the shear band changed to be amorphous at different focus planes. The entire region of the shear band was confirmed to be amorphous by analyzing images of different focus planes. However, since the shear band is not perpendicular to the plane of the image, only part of the amorphous region can be seen in one focus plane and one image, the other part will show lattice fringes. We have added discussion of this additional piece of evidence in the caption of Fig. S11. As a side note, tilting the image out of the zone axis makes the fringes within the band disappear. However, in that case the fringes also disappear in the crystalline grain and therefore we do not discuss this in the manuscript.

Reviewer #3 (Remarks to the Author):

(1) It has been broadly accepted that dislocation sources and pile up are hardly expected to exist in nc materials because of an increasing percentage of grain boundary atoms and a grain boundary accommodation mechanism should be responsible for the plastic deformation. With increasing grain boundary volume, the soft region, macroscopic strength softening is expected. This phenomenon seems to be independent of metallic structures. Hence the softening observed in SmCo5 is not surprising.

The reviewer is correct that grain softening (inverse Hall-Petch effect) is expected when the grain size is small enough due to the fact that GBs become responsible for strain accommodation. This observation has been seen in a number of MD simulations for different metallic systems and in some experiments (primarily on fcc metals). However, regardless of whether inverse HP regime is observed, for most metals there is a regime of traditional grain size strengthening (often described well by the empirical HP relation). The two main (surprising) findings of our study are lack of traditional HP regime in SmCo₅ samples (which can be explained by lack of dislocation activity) and amorphous shear bands that accommodate plastic strain in the regime of larger grain sizes.

We agree with the reviewer that the occurrence of inverse HP relation in SmCo₅ is not as surprising and it was helpful to realize that this is how the reviewer understood our claims. We have now modified the introduction to de-emphasize the inverse HP relation to state what is novel about our study. Thank you for this comment.

(2) From the references of the paper, deformation induced shear localization, which becomes the primary plasticity carrier, has been documented in literature, see ref. 24,25,26,and 34. It would be important to demonstrate atomic details of the amorphization process as a resultant of shear localization.

It is true that deformation induced shear localization is known in literature. However, it is a general term that captures many different mechanisms that are localized. We have discussed it in detail in lines 163-181. Briefly, in metallic systems it refers to localization of dislocations to a certain region. In SmCo₅ the process does not involve dislocations but instead it is a process of direct amorphization on non-crystallographic shear planes. A similar phenomenon was observed in ceramics (currently Refs. 36-40). However, in ceramics shear bands are precursor to fracture and the main mechanisms of brittle failure (also confirmed by MD simulations - Ref. 40). In contrast, the amorphous shear bands found in SmCo₅ can continuously accommodate strain without evolving into cracks, which has not been reported in literature. As to ref. 34 (currently Ref. 51), we would like to say that it has nothing to do with shear bands. We cited this work because we adopted the technique given in that reference to analyze the inverse FFT image.

We agree with the reviewer that the details of amorphization process on quasi-2D shear planes will be interesting to elucidate. We have already clarified some of these processes in our paper. For instance, in the case of boron carbide, there are still conflicting reports on the questions of whether shear bands nucleate on crystallographic planes and whether they involve a dislocation. We answered these specific questions for the case of SmCo₅. However, elucidating additional atomic-level details of amorphization process is beyond the scope of the current paper. Our paper is already substantial as we discover and provide detailed evidence (from simulations and experiments) for new deformation mechanisms and we discuss implications of this finding for grain size engineering. We would like to point out that, for instance, in the case of bulk metallic glasses both the criteria for glass stability as well as details of deformation mechanisms continue to be subject of research and the details are being debated. We are working on analyzing our simulations and experiments further to bring insights into such details for the case of intermetallics and we also hope that our findings will instigate future studies on this topic from other groups.

(3) It is important to show whether such shear bands exhibit grain size dependent behavior, e.g., structure, width, ... That could make the paper stands out from existing research reports.

We thank the reviewer for reminding us of this discussion. Actually, neither the width nor the structure of the shear bands shows grain size dependence. Our simulation shows that the width of a shear band is approximately 2 nm thick (irrespective of the grain size). The thickness of a shear band has been verified by the TEM observation, as shown in Fig. 3. We have also compared pair distribution functions (PDFs) of the shear bands found for different grain sizes and they are also independent of the grain size (within

numerical noise). We have added this information in the revised manuscript (lines 157-158 and 170-171).

There are further minor issues to be addressed.

(4) In presenting the mechanical behavior of SmCo5 as a function of grain size and made a comparison with Cu, the deformation mode for both simulations and experiments, tension/compression, constant volume/constant pressure..., should be specified clearly.

All the details of deformation simulations are given in the section “Simulation details” of the Supplementary Material. For the experimental details, they are specified in the corresponding sections of the Supplementary Material. We have also introduced most important details in the caption of Fig. 1 and Fig. S5.

(5) Many of strongly related references are missing. For instance, in both BCC and HCP metals, both traditional HP relation and the inverse one were reported.

grain boundary deformation has been documented in many pioneer work, from MD, Experiments, FEM simulations and so on:

- (a) Schiotz, J., Di Tolla, F.D. & Jacobsen, K.W. Softening of nanocrystalline metals at very small grain sizes. Nature 391, 561–563 (1998).*
- (b) Van Swygenhoven, H., Spaczer, M., Caro, A. & Farkas, Competing plastic deformation mechanisms in nanophase metals. Phys. Rev. B 60, 22–25 (1999).*
- (c) Shan, Z.W. et al. Grain boundary-mediated plasticity in nanocrystalline nickel. Science 305, 654–657 (2004).*
- (d) Wei, Anand. Grain-boundary sliding and separation in polycrystalline metals: application to nanocrystalline f.c.c. metals, Journal of Mechanics and Physics of Solids, 52, 2587-2616 (2004).*

We have added these and other additional citations in the revised manuscript. With respect to the references about GB activity recommended by the reviewer [we labeled them as (a-d)], Refs. (a) and (b) reported GB sliding as the dominant strain accommodation mechanism in nanocrystalline Cu and Ni as determined from MD simulations. We have thus cited them in the introduction part of the revised manuscript (Refs. 12 and 13). Ref. (c) reported the competition between deformation controlled by nucleation and motion of dislocations and that controlled by diffusion-assisted GB activity in nanocrystalline Ni. GB diffusion is not the focus of our paper, but it added the citation when we introduced our nanoindentation test with different strain rates [see the response to the comment (6)]. Ref. (d) modeled the competition between deformation due to GBs sliding and that arising from grain interiors, which has been cited when we discuss the grain size dependence of strain accommodation mechanism (Ref. 43).

(6) In addition, is grain boundary diffusion an important mechanism for plasticity in SmCo₅ at sufficiently small grains? Such mechanism seems to be important for FCC metals, see Gao et al. Enhanced strain-rate sensitivity in f.c.c. nanocrystals due to grain-boundary diffusion and sliding, *Acta Materialia*, 56, 2008, 1741-1752. A possible way to examine is to using the indentation tests at different loading rate to check the rate-sensitivity and also using dwelling step to see stress relaxation.

Per the reviewer's recommendation, we have carried out additional nanoindentation tests at different strain rates and also tests with different dwelling time. The results are shown in the figures below and are shown in Fig. S14 of the Supplementary Material. We can see that the maximum displacement of the indenter slightly increases by about 2 nm when the dwelling time increases from 10s to 200s. After then, the maximum displacement is stabilized. Besides, the maximum displacement of the indenter shows no noticeable dependence on the loading rate. These results indicate that GB diffusion may not be significant in SmCo₅ according to the investigation in the reference suggested by the reviewer. We have added this information in our manuscript with the reference cited (lines 233-235).

Fig. S14| Nanoindentation test of a strain-rate dependence (a) The maximum displacement of the indenter as a function of a dwelling time under a force of 6000 μN. (b) The maximum displacement of the indenter as a function of loading rate with a maximum force of 6000 μN. Error bars represent the standard deviation of the mean from the average values of the displacements. Experiments were carried out on a SmCo₅ sample with a 12 nm average grain diameter.

REVIEWERS' COMMENTS:

Reviewer #1 (Remarks to the Author):

Reviewer#1 (Remarks to the Authors):

Authors of the manuscript had performed additional experiments and confirmed the large plasticity by micropillars compression tests. This is a very insightful result, showing amorphization by plastic shearing in an intermetallic compound. I am satisfied with the text revision in the manuscript. In the form submitted is worthy of being published in Nature communications.

Reviewer #2 (Remarks to the Author):

The authors have gone to great lengths to revise their manuscript by adding extensive additional experimental work and replying/reacting to all comments raised in great detail.

I recommend this manuscript for publication in Nature Communications.

Reviewer #3 (Remarks to the Author):

The authors largely addressed my concerns in the first round. I am still conservative regarding the impact of the work to people working on plasticity of crystalline metals. From this aspect, I also disagree with the comments by other reviewers regarding its importance for plasticity. The observed amorphization from MD system would soon lead to localization and failure in real materials, as we commonly see in metallic glasses which tiny shear bands govern their plasticity. If it really contributes to plasticity and is able to retard localization, a simulation on a single crystal is much better for demonstration.

Other minor issues: the authors used different terms including crystalline metals, metals, and polycrystalline intermetallic. Those terms are quite different. Please be specific.

Comment from reviewer #3

The authors largely addressed my concerns in the first round. I am still conservative regarding the impact of the work to people working on plasticity of crystalline metals. From this aspect, I also disagree with the comments by other reviewers regarding its importance for plasticity. The observed amorphization from MD system would soon lead to localization and failure in real materials, as we commonly see in metallic glasses which tiny shear bands govern their plasticity. If it really contributes to plasticity and is able to retard localization, a simulation on a single crystal is much better for demonstration.

Other minor issues: the authors used different terms including crystalline metals, metals, and polycrystalline intermetallic. Those terms are quite different. Please be specific.

We thank the reviewer for the additional suggestion. We have done tensile deformation of a single crystal with the true strain up to 20%. In Supplementary Figure 9, we show that the single crystal can be deformed plastically assisted by the multiple amorphous shear bands, consistent with our discussions and conclusions.

The difference between the amorphous shear bands here and those found in metallic glasses is that the latter results in cracks due to generation of free volume [see Mater. Sci. Eng. R: Reports 74, 71-132 (2013)], while the amorphous shear bands here bring about a free volume that is one order of magnitude smaller.

In our manuscript, we compared our work with those reported in other metals. This is a natural comparison, as for instance the dislocation plasticity was mainly investigated in metallic systems and also the conventional shear bands were reported in polycrystalline metals. We have carefully examined the terminology and now there is no ambiguous description. We should also keep in mind that intermetallics are still (special) metallic system, so such comparison makes good sense.